# Regulating the glucose-6-phosphate dehydrogenase encoding gene *gsdA* and its impact on growth and citric acid production in *Aspergillus niger*

Susanne Fritsche[1,2], Valeria Ellena[1,2], Güler Demirbas-Uzel[2,3], Matthias G. Steiger[1,2,3]*

**1** acib - Austrian Centre of Industrial Biotechnology, Vienna, Austria, **2** Institute of Chemical, Environmental and Bioscience Engineering, Research Group Biochemistry, Technische Universität Wien, Vienna, Austria, **3** Christian Doppler Laboratory for Sustainable Bioproduction with Fungal Systems through Targeted Strain Development, Institute of Chemical, Environmental and Bioscience Engineering, Research Group Biochemistry, Technische Universität Wien, Vienna, Austria

* matthias.steiger@tuwien.ac.at

## Abstract

Glycolysis in *A. niger*, a key organism in industrial biotechnology, provides essential precursors for efficient citric acid production. Glucose-6-phosphate dehydrogenase (G6PD), encoded by the gene *gsdA,* is a critical point in the cellular metabolism as it determines the metabolic fate of glucose-6-phosphate by redirecting it into the pentose phosphate pathway (PPP). Despite its decisive position in the metabolic network the functional role of G6PD and its impact on citric acid synthesis and growth is not fully understood. Here, we present an *A. niger* strain expressing a ptet-on regulated version of *gsdA* at the *pyrG* locus. The native gene was disrupted and hence, *gsdA* expression was based on a single copy level. Under non-inducing conditions, the strain was not growing on glucose. On gluconate, a precursor for an intermediate of the oxidative PPP, growth was restored but delayed compared to the control strain expressing *gsdA* under the native promoter. Furthermore, citric acid production was monitored in dependency of different *gsdA* induction levels using doxycycline. At low induction levels, the yield on glucose was enhanced by 49% compared to the control strain, albeit with reduced growth leading to lower titers. Through supplementation of the medium with gluconate, we anticipated to provide precursors for biomass production for efficient metabolization of glucose to citric acid. Without the native regulation of the *gsdA* gene growth and citric acid production were time delayed. However, the yield of the *gsdA*-regulated strain was higher compared to the control after 120 h of cultivation and was positively influenced with an increasing proportion of gluconate in the medium. The findings of this study underscore the dependency of growth and citric acid production on *gsdA* expression in *A. niger*.

**Data availability statement:** The plasmids listed in Table 1 were submitted to Addgene (https://www.addgene.org/) and are now publicly available. The identifiers are listed in Table 1 and can be used to directly approach the respective site. Plasmid: 222587 (https://www.addgene.org/222587/). All other data is directly available in the manuscript or in the supporting information submitted.

**Funding:** MGS received funding from the COMET center: acib: Next Generation Bioproduction which is funded by BMK (Ministry of Climate Action and Energy), BMDW (Ministry for Digital and Economic Affairs), SFG (Steirische Wirtschaftsförderungsgesellschaft), Standortagentur Tirol, Government of Lower Austria and Vienna Business Agency in the framework of COMET - Competence Centers for Excellent Technologies. The COMET-Funding Program is managed by the Austrian Research Promotion Agency FFG. MGS received funding from the Christian Doppler Laboratory for Sustainable Bioproduction with Fungal Systems through Targeted Strain Development by the Austrian Federal Ministry of Labour and Economy, the National Foundation for Research, Technology and Development, the Christian Doppler Research Association, and Jungbunzlauer Austria AG. The funding agencies encourage the publication of scientific results but did not influence the study design, data collection, and analysis, decision to publish, nor preparation of the manuscript.

**Competing interests:** The authors have declared that no competing interests exist.

## Introduction

*Aspergillus niger* is a filamentous fungus renowned for its capacity to produce and secrete large quantities of industrially relevant chemicals. Its production capabilities have been extensively harnessed to manufacture enzymes, such as amylases and proteases. Moreover, *A. niger* is a cell factory for gluconic and oxalic acid production and a key producer of citric acid, the world's second largest consumed organic acid [1–3]. The continuous importance of *A. niger* in industrial biotechnology underpins the ongoing effort to enhance the organism's production capability by understanding metabolic routes and engineering of selected enzymatic conversions [4,5]. The glucose metabolism shown in Fig 1 includes glycolysis, the pentose phosphate pathway (PPP) and the tricarboxylic acid (TCA) cycle and is a target for such modifications. Due to its central role in providing NADPH for anabolism, studies have focused on the development of PPP to improve the production of proteins and valuable chemicals in several industrially relevant organisms [6,7]. In *K. phaffii,* overexpression of NADPH-generating reactions was reported to increase recombinant protein production [8,9]. In *S. cerevisiae*, engineered NADPH regeneration improved pentose fermentation and production of ethanol from xylitol by 50% [10]. By overexpressing enzymes of the PPP, NADPH dependent production of glycoamylase was shown to be enhanced in *A. niger* [11]. The oxidative part of the PPP (oxPPP) harbors two NADPH generating enzymatic steps. Glucose-6-phosphate dehydrogenase (G6PD) is the first and rate-limiting enzyme of the oxPPP and converts glucose-6-phosphate to 6-phosphogluconolactone leading to the reduction of NADP+ to NADPH. Subsequently, 6-phosphogluconolactone is hydrolyzed to 6-phosphogluconate. The latter can also derive from gluconate. In *A. niger* gluconate is produced from glucose by glucose oxidase which is located at the cell wall but is also found in the cytoplasm [12]. It can enter the oxPPP via phosphorylation by a gluconokinase leading to 6-phosphogluconate [13]. The second NADPH generating reaction of the oxPPP is the conversion of 6-phosphogluconate to ribulose-5-phosphate catalyzed by 6-phosphogluconate dehydrogenase (6PGD). Together with the reaction of G6PD two NADPH molecules are produced and are crucial cofactors for reductive biosynthesis, biomass production [14,15] and maintaining the cellular redox balance [16,17]. Ribulose-5 phosphate is then channeled into the non-oxidative part of the PPP (non-oxPPP). This part primarily deals with interconversion of sugar phosphates. The 3-carbon to 7-carbon sugars are essential for various biosynthetic processes that can enter glycolysis or are used in nucleotide and amino acid biosynthesis to build nucleic acid and proteins, respectively.

G6PD of the oxPPP is a ubiquitous enzyme and therefore generally accepted as a housekeeping gene. The active enzyme G6PD is in a dimer – tetramer equilibrium [18], was purified from *A. niger*, *A. nidulans* and *A. oryzae* and was biochemically characterized [19,20]. The homologous gene *ZWF1* in *S. cerevisiae* has been studied in more detail to uncover its functional role within the metabolism. Overexpression of *ZWF1* was linked to enhanced NADPH concentrations [21] and a strain lacking G6PD was viable on glucose but more sensitive to oxidative stress [22]. The deletion of *ZWF1* was compensated by an aldehyde dehydrogenase encoded by *ALD6* or also by isocitrate dehydrogenase for alternative NADPH production. A Δ*ZWF1/ALD6* strain was, however, not viable [23,24]. In *A. niger*, overexpression of *gsdA* was studied and resulted in increased G6PD activity and biomass production. Interestingly, the intracellular NADPH pool remained at wild-type levels [11,25].

In this study the dependency of *A. niger* on G6PD is evaluated through controlled *gsdA* expression on a single copy level. We performed phenotypic characterization of the *gsdA*-regulated strain on different media. While the PPP is commonly engineered for protein production, we here focus on the influence of G6PD on citric acid formation.

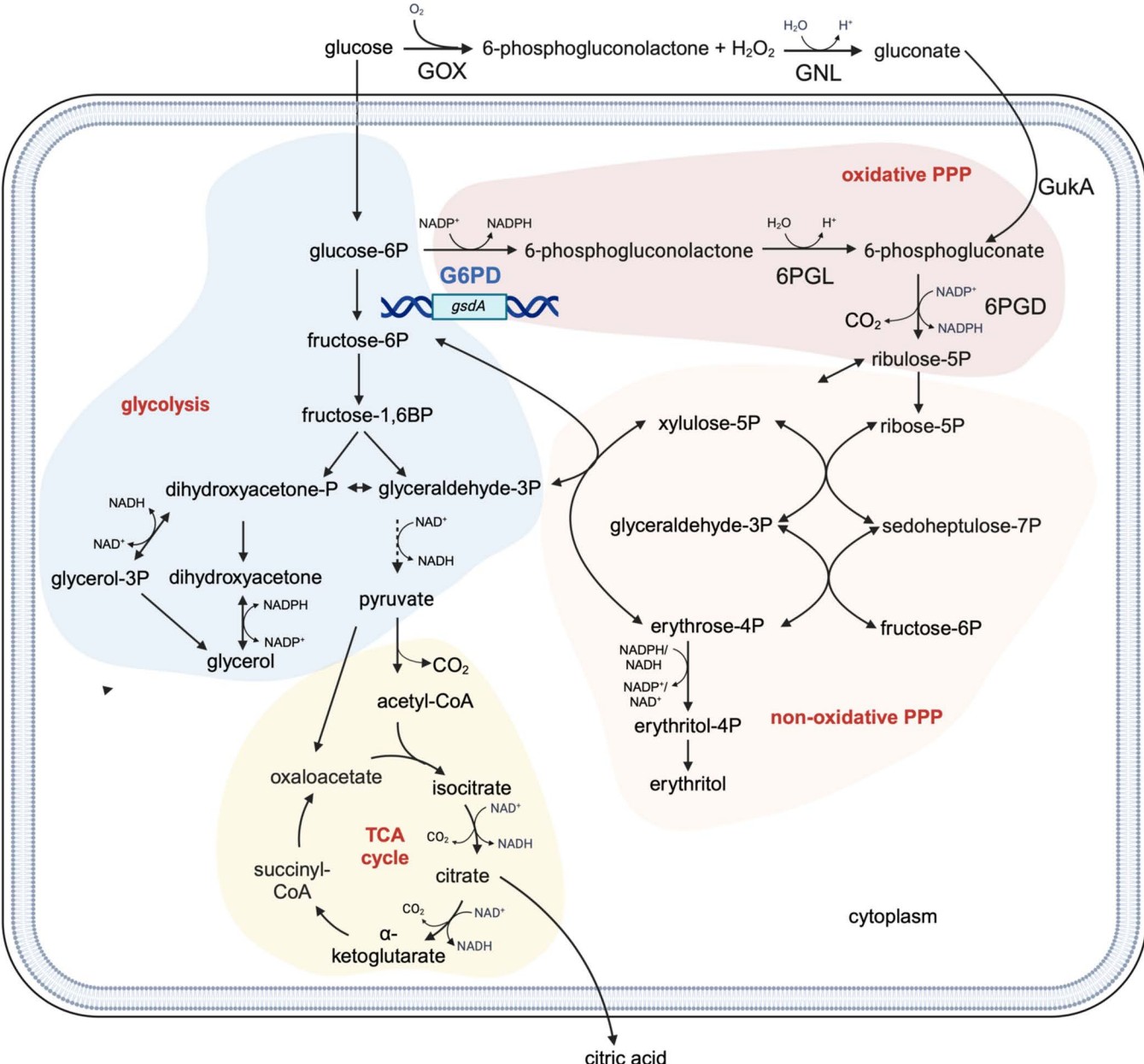

**Fig 1. Production of citric acid is studied under up- and downregulation of the PPP via the G6PD encoding gene *gsdA*.** Glucose is uptaken by the fungal cell and metabolized via glycolysis (blue) to pyruvate that enters the TCA cycle (yellow) via acetyl-CoA. The metabolite citrate is produced and secreted as citric acid. Glycolysis is linked to the oxidative PPP (red) via the NADPH-generating reaction of G6PD and further downstream to the non-oxidative PPP (green). Abbreviations: GOX= glucose oxidase; GNL= gluconolactonase; GukA= gluconokinase; PPP= pentose phosphate pathway; P= phosphate; G6PD= Glucose-6-phosphate dehydrogenase; 6PGL= 6-phosphogluconolactonase; 6PDG= 6-phospholuconate dehydrogenase; CoA= Coenzyme A; TCA= Tricarboxylic acid. Dashed arrow means more reactions in between. Enzymes of the oxPPP are highlighted in blue. Created in BioRender. Fritsche, S. (2024) BioRender.com/c53c134.

## Materials and methods

### Construction of plasmids

Integrative and Cas9/sgRNA encoding plasmids used in this study are listed in Table 1. Plasmid construction was done with the Golden Gate cloning system [26,27] following the

**Table 1. List of plasmids used throughout the study.**

| Identifier | Name | Relevant cassette or insert | Function | Reference Addgene |
|---|---|---|---|---|
| pSF503 | BB3_gsdA_OE | *ptet-on:gsdA_mutPAM:ttrpC* | Construction of SF369 | 222587 |
| pSF522 | BB3_ZWF1_OE | *ptet-on:ZWF1:ttrpC* | Construction of SF274 | 222588 |
| pMST620 | BB3_gpyrG2 _cas9 | *pmbfA:gRNA2(pyrG):ttrpC* | Construction of SF369, SF274 | 90277 |
| pMST621 | BB3_gpyrG1 _cas9 | *pmbfA:gRNA1(pyrG):ttrpC* | Construction of SF387, SF388 | 90276 |
| pSF511 | BB3_ggsdA1_cas9 | *pmbfA:gRNA1(gsdA):ttrpC* | Construction of SF283, SF284, SF285, SF395 | 222719 |
| pSF2 | BB_gsdA_disruption_ cassette | 5'-*gsdA:pyrG*(A.*nidulans*):3'-*gsdA* | Construction of SF395 | 222590 |
| pSF8 | BB2_tet-on_gsdA | *ptet-on:gsdA_mutPAM::ttrpC* | Construction of pSF503 | 222592 |
| pSF18 | BB2_tet-on_ZWF1 | *ptet-on:ZWF1:ttrpC* | Construction of pSF522 | 222593 |
| BB2_6 | L_AB_syn_BbsI | Spacer | Vector to obtain SF374 and pSF8, pSF18 | 89917 |
| pMST1212 | BB2_E_BC_pcoxA:pyrGtrunc_ AMA1_2.8 | *pcoxA:pyrG*$^{m2,trunc}$ | Split *pyrG* marker for homologous integration | 90286 |
| pMST635 | BB3_L_AC_hyg_ KanOri_pyrG | 5'-*pyrG* | 5'-*pyrG* sequence for homologous integration at *pyrG* locus | 90280 |

protocol of Sarkari and colleagues [28] for gene integration and expression at the *pyrG* locus of *A. niger* strains.

Primers used for the construction of plasmids and to generate strains are listed in Table 2. Generally, the expression cassettes harbored an inducible ptet-on promoter system regulated by addition of doxycycline to the cultivation medium [29] and a *trpC* terminator (*ttrpC*) sequence. The coding sequence of *ZWF1* from *Komagataella phaffii* GS115 was obtained from Nocon and colleagues [9] to construct pSF18 and pSF522. For the CDS *gsdA_mutPAM* in pSF8 and pSF503 from *A. niger* ATCC 1015 the ORF of *gsdA* (NCBI reference number: XM_001400305.3) was amplified via PCR using genomic DNA using primers P1 and P2. *BsaI* recognition sites were removed via overlapping PCRs using primers P3/4 and P5/6. The PAM site for gRNA1(*gsdA*) expressed with pSF511 was mutated (5'-GGG to 5'-AGG) using primer P7 and P8 to only target the native *gsdA* locus. The final nucleotide sequence of the PCR product is given in S1 File. gRNA sequences with underlined PAM sites in pMST621, pMST620, pSF511 and pSF539 were 5'-ATCCCAATGCTCTGGCGAAG<u>AGG</u>, 5'-GTAGGTCAATTGCGACTTGG<u>AGG</u>, ACGGTGGTGAAAACGCTGGG<u>GGG</u> and 5'-ATCTCTCCTAGACCTTCTCC<u>AGG</u>, respectively and selected using CHOPCHOP [30]. pSF2 encodes a disruption cassette to knock out the native *gsdA* gene by integration of a marker gene via a homologous recombination event. For this the *pyrG* gene from *A. nidulans* FGSCA4 (NCBI reference number: XM_658669.1) was amplified from genomic DNA including 766 and 1011 bp of its up- and downstream sequence using primer P9 and P10. *BsaI* and *BpiI* recognition sites were removed performing overlapping PCRs using primers P11/12 and P13/14. The 5' flank was a homologous sequence of 756 bp of *gsdA* amplified from genomic DNA using P15 and P16 and was selected seven bp upstream from the PAM site of gRNA1(*gsdA*). The 3' flank was a homologous sequence of 553 bp and amplified using P17 and P18. The fragments were joined using overlapping PCR and cloned into the vector resulting in pSF2. The disruption cassette in pSF2 was then linearized by restriction digest with *BpiI* (New England Biolabs Ipswich, MA) and purified from a gel using HiYield PCR Clean-up/Gel Extraction Kit following the manufacture's protocol. The expression cassette to

**Table 2. List of primer sequences used throughout the study.**

| Primer | Name | Sequence (5'-3') |
|---|---|---|
| P1 | FS2_CDS gsdA mutPAM_fwd | GGTCTCGCATGGCCAGCACAATAGCACGCACTGAGGAACGCCAGAATGCTGGGTGAGTTTTGCGGTCTGCCTCTCTGACATCAC |
| P2 | FS3_CDS gsdA mutPAM_rev | ATGGTCTCCAAGCTTACAGACGGTTGGGGGTGGAAGTCAAGGGCCACTGG |
| P3 | Bsal_mut1_fwd | GAGATCTTGCAAAGAAGAAAACCGTCAGTGACGACC |
| P4 | Bsal mut1_rev | GGTCGTCACTGACGGTTTTCTTCTTTGCAAGATCTC |
| P5 | Bsal_mut2_fwd | GCCTACAAGGAGGACGAAACCGTTCCCCAGGATTCC |
| P6 | Bsal_mut2_rev | GGAATCCTGGGGAACGGTTTCGTCCTCCTTGTAGGC |
| P7 | gsdA_mutPAM_fwd | GAATCTACTACATGGCCCTCCCTCCCAGCGTTTTCACCACCGTTTCC |
| P8 | gsdA_mutPAM_rev | GGAAACGGTGGTGAAAACGCTGGGaGGGAGGGCCATGTAGTAGATTC |
| P9 | pyrG_A.nidulans_fwd | ATGGTCTCCCATGATAACTTCGTATAGCATACATTATACGAAGTTATCCCATGCTAATCATATAA |
| P10 | pyrG_A.nidulans_rev | GTGGTCTCCAAGCCATAACTTCGTATAATGTATGCTATACGAAGTTATCAGAGCTCTAATTCTAGAGAC |
| P11 | pyrG_A.nidulans_Bpil1_fwd | CGCCATCATGTCGTCGAAGTCC |
| P12 | pyrG_A.nidulans_Bpil_rev | GGACTTCGACGACATGATGGCGGTTCTCCAATGATT |
| P13 | pyrG_A.nidulans_Bsal_fwd | TGACTTTAAAGACGCGAATCAACGAGGCCTCCTGATTCT |
| P14 | pyrG_A.nidulans_Bsal_rev | GAGGCCTCGTTGATTCGCGTCTTTAAAGTCA |
| P15 | FS1_gsdA 5' flank_fwd | ATGGTCTCCGGAGAGACGTGAGGATTCACCACC |
| P16 | gsdA 5' flank_pyrG overhang_rev | ATGCTATACGAAGTTATCATATGTAGTAGATTCTGTTCTGCTCC |
| P17 | gsdA 3' flank_pyrG overhang_fwd | GCATACATTATAACGAAGTTATG AGGAGATCATCCCCATGGAATACC |
| P18 | gsdA 3' flank_FS2_rev | GTGGTCTCCCATGAGACCAAAGACACTAAGGCTCA |
| P19 | pyrG_5' out_fwd | TTTTGGTTAGCACCTACGCTAGTCTATCAG |
| P20 | pyrG_3' out_rev | CATCGGAAGCACAATGAGGCGAGTTT |
| P21 | gsdA_ch_1_fwd | TACATCAAGACCCCTACCAAGG |
| P22 | gsdA_ch_1_rev | CCAGGATTGACTCACGATGATA |
| P23 | pyrG_ch_1_fwd | TCTGGATTTACGAATCAGGGTC |
| P24 | pyrG_ch_1_rev | GCTCCTTAGTGGTGGTAACGTC |
| P25 | 5'gsdA_fwd | CTGTGACATTATATTCGTCAAGCTATAG |
| P26 | pyrG_A.nidulans_rev | CTTGTACTCACGAATCCCATCACA |
| P27 | pyrG_A.nidulans_fwd | CAAGTCTCCTGACTTTAAAGACGCG |
| P28 | 3'gsdA_rev | TCAGGTACCTCGTCTTATCG |

be integrated at the *pyrG* locus for generating the control strain was a spacer sequence from the empty linker BB2_L_AB_syn_Bbsl (5'- CGCAAAAAACCCCGCCCCTGACAGGGCGGGGTTTTTTTCGC). For transformation, the linker was cloned together with the *pyrG* split marker from pMST635 into the recipient vector pMST1212.

Proliferation of all plasmids was done in *E. coli* Top10. Transformants were grown on LB [31] supplemented with 50 μg/mL kanamycin, 100 μg/mL ampicillin or 100 μg/mL hygromycin B.

## Strains and media

All transformants derived from wild-type *Aspergillus niger* ATCC 1015 and are listed in Table 3. *A. niger* strains were maintained on minimal medium (MM) containing 1% glucose, 1X ASPA+N, 0.002 M MgSO$_4$, 1X trace element solution and 1.5% agar as previously described [32]. *PyrG* deficient (*pyrG-*) strains were selected for uridine auxotrophy on medium containing 5-FOA (2g/L) and were maintained with supplementation of 10 mM uridine. 1 μg/mL doxycycline was added to the medium to maintain strain SF395. Strains were grown for 7 days at 30°C before harvesting. For this, conidia were suspended with 0.1% Tween 20 solution and filtrated through Miracloth. Conidia were then centrifuged at 5000 rpm for 10 min and resuspended in 0.1% Tween 20 and this step was done twice.

**Table 3. A.** *niger* **strains used or generated in this study.**

| Identifier | Name | Background strain | Relevant genotype | Reference |
|---|---|---|---|---|
| WT | wild-type | | ATCC 1015 | [33] |
| A621 | *pyrG*ᵐ¹ | WT | *pyrG-* | [34] |
| SF274 | *ZWF1* OE | A621 | *pyrG::ptet-on:ZWF1:ttrpC* | This study |
| SF369 | *gsdA* OE | A621 | *pyrG::ptet-on:gsdA_mutPAM:ttrpC* | This study |
| SF374 | control | A621 | *pyrG::spacer* | This study |
| SF387 | *gsdA* OE_pyrG- | SF369 | *pyrG::ptet-on:gsdA_mutPAM:ttrpC; pyrG-* | This study |
| SF388 | *ZWF1* OE_pyrG- | SF274 | *pyrG::ptet-on:ZWF1:ttrpC; pyrG-* | This study |
| SF283 | *ZWF1* OE_*gsdA*mut_1 | SF388 | *pyrG::ptet-on:ZWF1:trpC;* XM_001400305.3:p.Ser183Thr | This study |
| SF284 | *ZWF1* OE_gsdAmut_2 | SF388 | *pyrG::ptet-on:ZWF1:trpC;* XM_001400305.3:pSer183Asp | This study |
| SF285 | *ZWF1* OE_gsdAmut_3 | SF388 | *pyrG::ptet-on:ZWF1:trpC* XM_001400305.3:pSer183His | This study |
| SF395 | *gsdA*-regulated strain | SF387 | *pyrG::ptet-on:gsdA_mutPAM:ttrpC; gsdA::pyrG(A. nidulans)* | This study |

## Transformation of *A. niger*

*A. niger* protoplast transformations were performed according to the previously described protocol [32] using 0.4 mg/mL VinoTaste (Novozymes, Bagsværd, Denmark) as lysing enzymes dissolved in a buffer containing sorbitol, MES and $CaCl_2$ (SMC). Transformants were selected on MM plates containing 200 μg/mL hygromycin B and purified by three rounds of single colony isolation on MM with hygromycin B (100 μg/mL). This was followed by two rounds on MM. All media contained 10 mM uridine or 1 μg/mL doxycycline when applicable.

## Genetic manipulation using CRISPR/Cas9

Genomic integration was performed according to the protocol reported by Sarkari et al., 2017 [28]. In brief, background strain A621 is *pyrG* deficient (*pyrG-*) and a functional *pyrG* locus is obtained by a homologous recombination event of an expression cassette that is flanked with a *pyrG* split marker. This enables selection on the integration event via uridine prototrophy. The integration of expression cassettes pSF522 and pSF503 in SF274 and SF369, respectively, was verified by PCR after genomic DNA extraction using primers P19 and P20 and a Q5 Polymerase (New England Biolabs, Ipswich, Massachusetts, USA). Strains SF274 and SF369 were then transformed with 1 μg of the Cas9/gRNA plasmid pSF511 to achieve deletion of the native *gsdA* gene. The target locus *gsdA1* was sequenced with primers P21 and P22. Further-more, transformation of SF274 and SF369 was done with 1 μg of pMST621 to obtain *pyrG* deficient strains SF387 and SF388. PCR of the manipulation site was performed using primers P23 and P24 to verify the deletion event. To disrupt the native *gsdA* gene in strains SF387 and SF388 1 μg of the Cas9/gRNA plasmid pSF511 was co-transformed with 400 ng of the disruption cassette derived from pS2. Integration of the disruption cassette was confirmed by PCR and subsequently by Sanger sequencing (Microsynth, Balgach, Switzerland) with primers P25 to P28.

## Phenotypic characterization

A dilution series of conidia in 0.1% Tween 20 was prepared with concentrations of 5 x $10^5$, 5 x $10^4$, 5 x $10^3$, 5 x$10^2$, 5 x$10^1$ conidia/mL. For growth tests, MM was supplemented with 1% glucose or equimolar amounts of Na-gluconate. For acidification test on $CaCO_3$-plates the

medium was prepared as following: 30 mL of Medium 1 (5 g/L $CaCO_3$ and 12 g/L noble agar) was poured as the bottom layer. 30 mL of Medium 2 was poured on top and consisted of solution A (0.374 g/L $KH_2PO_4$, 0.374 g/L $MgSO_4 \times 7\,H_2O$, 0.065 g/L $CaCl_2 \times 2\,H_2O$, 0.093 g/L citric acid anhydrous, 220 g/L glucose monohydrate, 0.004 g/L $FeSO_4 \times 7\,H_2O$, 0.009 g/L $CuSO_4 \times 5\,H_2O$, 0.006 g/L Zn-citrate, 0.121 g/L $K_2SO_4$, pH adjusted to 3.5 with NaOH), solution B (24 g/L noble agar) and 25 ml/L of an N-Stock (35.52 g/L $(NH_4)_2SO_4$, 0.374 Urea). To prepare Medium 2 solution A (weighted in for 1L and dissolved in 0.475 L) was mixed with B (0.5 L) and 25 mL of the N-Stock to obtain in total 1 L.

2 µL of each suspension and strain was applied to test plates and incubated for 90 h. All spotting tests were performed with 0, 1, 2.5 and 5 µg/mL doxycycline.

## Cultivation

All cultivations were performed in Vogel's medium (VM; 2.5 g/L $Na_3$-citrate $2 \times H_2O$, 5 g/L $KH_2PO_4$, 2 g/L $NH_4NO_3$, 0.2 g/L $MgSO_4\,7 \times H_2O$, 0.1 g/L $CaCl_2\,2 \times H_2O$, 0.5 mg/L citric acid $1 \times H_2O$, 0.5 mg/L $ZnSO_4\,7 \times H_2O$, 0.1 mg/L $Fe(NH_4)_2\,(SO_4)_2\,6 \times H_2O$, 0.025 mg/L $CuSO_4\,5 \times H_2O$, 0.005 mg/L $MnSO_4\,1 \times H_2O$, 0.005 mg/L $H_3BO_3$, 0.005 mg/L $Na_2MoO_4\,2 \times H_2O$, and 0.005 mg/L biotin) [35]. Determination of citric acid, erythritol, glycerol and D-glucose or gluconate was performed as previously described [36] on a HPLC (Shimadzu, Kyoto; Japan) equipped with a Aminex HPX-87 H (300×7.8 mm, BioRad, Hercules, CA). A refraction index detector (RID-10 A, Shimadzu) was used for detection of all compounds. The column was operated at 50°C, 0.6 mL/min flow rate, and 0.004 M $H_2SO_4$ as mobile phase. In all samples, citric acid, erythritol, glycerol or D-glucose were determined based on a calibration curve using pure standards.

## Microtiter plate assay

Cultivation was performed in a 96-well plate and VM supplemented with 200 g/L D-glucose and 0,1 and 3 µg/mL doxycycline in order to induce *ZWF1* or *gsdA* in the respective overexpression strains SF274 and SF369. For each condition, 10 mL of the medium were inoculated with $10^8$ conidia/L and 100 µL were taken for cultivation that was performed at 30°C, 70% humidity, and 240 rpm for 72 h in a Minitron incubation shaker (Infors HT Minitron, Bartelt, Austria, 5 cm amplitude). The cultivation was done in eight biological replicates and the supernatant was collected by filtration for HPLC analysis.

## Shake flask cultivation

VM was either supplemented with 200 g/L D-glucose or composed of 80 and 20% or 60 and 40% D-glucose and Na-gluconate, respectively. The medium was supplemented with 0, 0.25, 0.5, 1 and 2.5 µg/mL doxycycline in order to induce *gsdA* overexpression in the strain SF395. $10^9$ conidia/L were inoculated in 100 mL medium and grown at 30 °C and 200 rpm on a rotary shaker for 168 h. The cultivation was done in three biological replicates and analyzed by HPLC. The mycelium was harvested using Miracloth, and washed with sterilized distilled water, and mycelium, dried between Whatman paper, was used to prepare cell-free extracts for subsequent total protein determination and enzymatic activity experiments.

**Cell free protein extraction.** The harvested dry mycelium was weighed (~150 mg) and resuspended in a Protein Extraction Buffer (PEB) (50 mM potassium phosphate buffer, 1 mM dithiothreitol, 2 mM $MgSO_4$, pH 7.5) at a ratio of 1 g of mycelium per 3 mL of buffer. The cells were disrupted using a Fast-Prep-24 (MP Bio-medicals, Santa Ana, CA, USA) with 0.37 g of small glass beads (0.1 mm diameter), 0.24 g of medium glass beads (1 mm diameter), and a single large glass bead (5 mm diameter) at 6 m/s for 15 seconds, repeated 10 times, alternating

with cooling on ice. The resulting cell-free extract was clarified by centrifugation at $16,000 \times g$ for 15 minutes at 4°C, and the supernatant was carefully collected into 1.5 mL tubes. To ensure further clarification, the centrifugation step was repeated. All steps were performed on ice to prevent protein degradation [37].

**Bradford protein assay.** The concentration of total protein in the samples for both strains SF369 (*gsdA* OE) and SF374 (control) was determined by measuring the UV absorbance at 595 nm according to the Bradford protein assay (Bio-Rad Protein Assay, Biorad, Hercules US) [38].

**Enzymatic activity.** The enzymatic activity of Glucose-6-Phosphate Dehydrogenase (G6PD) was assessed using the G6PD Activity Assay Kit (Sigma-Aldrich, MAK451, St. Louis, Missouri, US) following the manufacturer's instructions [39–41]. Cell-free extracted A. *niger* samples (~150 mg) were prepared in 500 μL of PEB (pH 7.5) and centrifuged at $16,000 \times g$ for 15 minutes at 4°C to collect the supernatant and used for enzymatic assay.

For the assay, 20 μL of the cell-free extract or serial dilutions or calibrator was loaded into a 96-well plate in triplicate, and 80 μL of working reagent, prepared by combining assay buffer, substrate, NADP/MTT, and diaphorase, was added to each well. The plate was mixed, and absorbance at 565 nm was recorded immediately ($T_0$) and after a 15-minute incubation at room temperature ($T_{15}$). The change in absorbance ($\Delta A_{565}$) was calculated, and G6PD activity was determined by normalizing sample values to a supplied calibrator. Specific enzyme activity was expressed as units per milligram total protein (U/mg), where one unit corresponds to the conversion of 1 μmol of substrate per minute under the assay conditions.

## Results

### Construction of a *gsdA* overexpressing strain

Glucose is mainly catabolized via glycolysis or alternatively enters the oxidative PPP at the level of glucose-6 phosphate. The first enzyme of this pathway is G6PD encoded by the gene *gsdA* (NCBI reference number: XM_001400305.3). Here, we want to investigate the effect of a regulated *gsdA* gene on the citric acid production capability of *A. niger*. For this, the coding sequence of the *gsdA* gene was integrated at the *pyrG* locus of *A. niger* under the control of the doxycycline inducible ptet-on promoter. Only the control strain SF374 harbors a 40 bp spacer sequence at the *pyrG* locus. The strains were cultivated with glucose and the secreted metabolites as well as the G6PD activity of the strains were determined (Fig 2).

As expected, citric acid, erythritol and glycerol titers are not influenced in the control strain by increasing doxycycline concentrations (Fig 2). However, the induction of the *gsdA* OE cassette by doxycycline has a significant impact on the metabolites secreted:

Citric acid titers were significantly reduced by increasing doxycycline concentrations in the *gsdA* OE strain (Fig 2A). Consequently, the yield on glucose decreased from $0.19 \pm 0.01$ g/g for the *gsdA* OE strain without induction to $0.04 \pm 0.01$ g/g when *gsdA* was overexpressed with 1 μg/mL of doxycycline. Increased *gsdA* expression had also a negative effect on the titers of erythritol and glycerol. Titers were reduced from $1.05 \pm 0.03$ to $0.46 \pm 0.09$ for erythritol and $2.73 \pm 0.03$ to non-detectable amounts for glycerol when comparing data from the *gsdA* OE strain under non-inducing condition and induction with 1 μg/mL of doxycycline (Figs 2C and 2D).

As expected, the specific G6PD activity in the control strain as well as in the uninduced overexpression strain (SF369) remained at approximately $1.09 \pm 0.05$ U/mg. In the presence of doxycycline, the specific G6PD activity of the *gsdA* OE strain increased significantly, reaching $3.5 \pm 0.1$ U/mg.

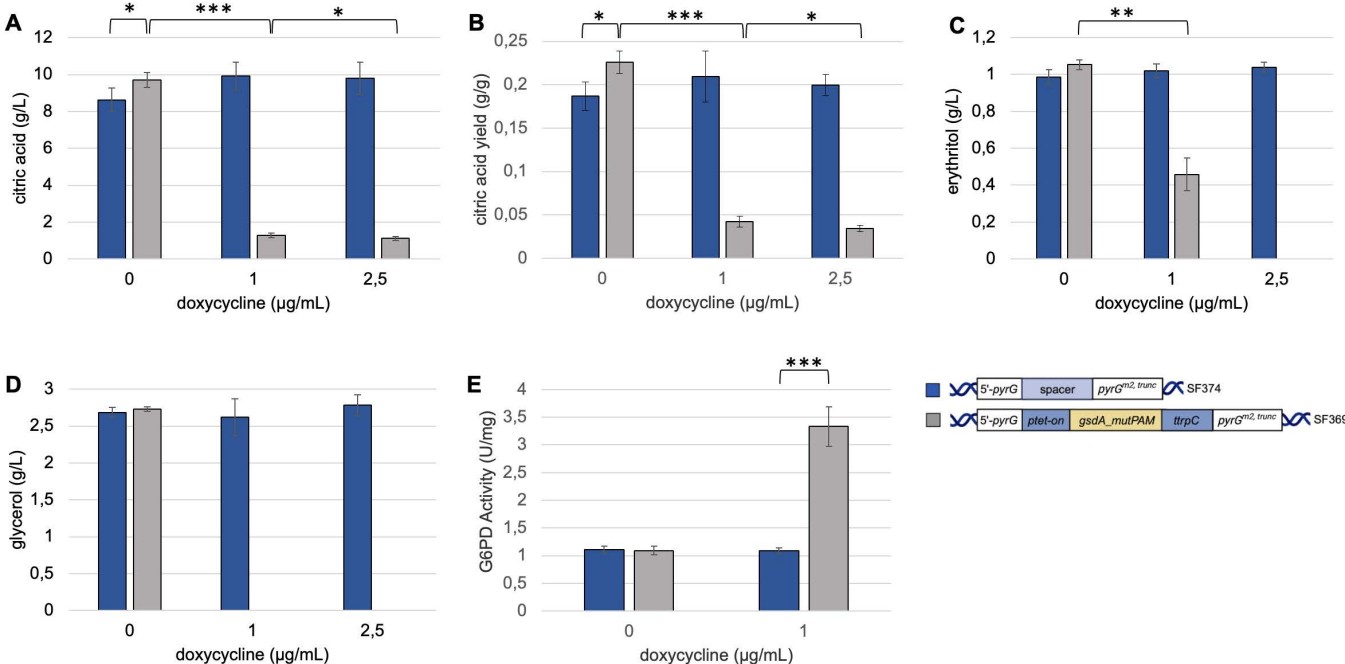

**Fig 2. Secreted metabolites and G6PD activity produced by *A. niger* overexpressing *gsdA*.** $10^8$ conidia/L of the control strain, SF374 (blue bars), or the gsdA overexpression strain, SF369 (grey bars), were inoculated in Vogel's medium supplemented with 0, 1 and 2.5 µg/mL doxycycline and cultivated in microtiter plates. Citric acid titers (A), citric acid yield (B), erythritol (C) and glycerol titers (D) of the cultivation supernatant were measured after 72 h of cultivation at 30°C. Cultivation was done with eight replicates and error bars indicate standard deviation. G6PD specific enzyme activity was determined in cell-free extracts from mycelia cultivated in shake flasks with Vogel's medium supplemented with 0, 1 µg/mL doxycycline and harvested after 72 hours in biological triplicates. Significant differences ($*p<0.05$; $**p<0.001$; $***p<0.0001$) were determined by t-test and highlighted by asterisks. Created in BioRender. Fritsche, S. (2024) BioRender.com/m29b218.

We showed that citric acid production is lowered when *gsdA* is overexpressed in *A. niger*. Conversely, we hypothesized that *gsdA* downregulation would shift the carbon flux towards glycolysis and thus should increase citric acid production. First, we tested the knock-out of the native *gsdA* gene with CRISPR/Cas9 using gRNA1(*gsdA*). However, no knock-out mutant could be recovered. Next, we tried to regulate the expression of *gsdA* at the native locus by replacing the promoter with a ptet-on promoter system but this was also not successful. Therefore, we continued to target the native *gsdA* gene copy in the *gsdA* OE strain SF369. The expression cassette with the PAM-mutated version of *gsdA* was induced during the transformation with the Cas9/gRNA encoding plasmid. This ensured *gsdA* gene expression during the knock-out attempt of the native copy. However, none of the transformants showed a mutation at the target site. To prevent the repair of the native *gsdA* with the second copy in the genome, we proceeded the knock-out approach with a strain harboring an expression cassette with the G6PD encoding gene *ZWF1* from *K. phaffii*.

Following this strategy, five clones were recovered with a nucleotide exchange at the target site of the gRNA1(*gsdA*). Nonsynonymous substitutions were found in three of the five clones on position XM_001400305.3:p.Ser138. Serine was changed to threonine (5'-AGC to 5'-ACC) in SF283, asparagine (5'-AGC to 5'-AAC) in SF284 or histidine (5'-AGC to 5'-CAC) in SF285. The remaining two clones had synonymous substitutions at this position. Overall, neither growth retardation nor any other phenotypic change was observed compared to the parental strain in all five cases.

## Disruption of the native *gsdA* gene

With the CRISPR/Cas9 approach, synonymous and nonsynonymous mutations were obtained but ultimately, a *gsdA* knock-out strain was not recovered under these conditions. We then continued to target the native *gsdA* gene in *gsdA* and *ZWF1* OE strains SF369 and SF274, respectively, using a selectable disruption cassette. First, the strains SF369 and SF274 were mutated at the native *pyrG* site using CRISPR/Cas9. The deletion of 27 bp was confirmed via sequencing (S2 File) leading to uridine auxotrophy and strains SF387 and SF388 (Fig 3A). The disruption cassette harbored the CDS of *pyrG* from *A. nidulans* (NCBI reference number: XM_658669.1) including approximately 1 kb of its up- and downstream region. The sequence was flanked with homologous arms (~ 0.5 kb) of the native *gsdA* gene of *A. niger*. During co-transformation of the linearized DNA with the CRISPR/Cas9 expression plasmid, the OE

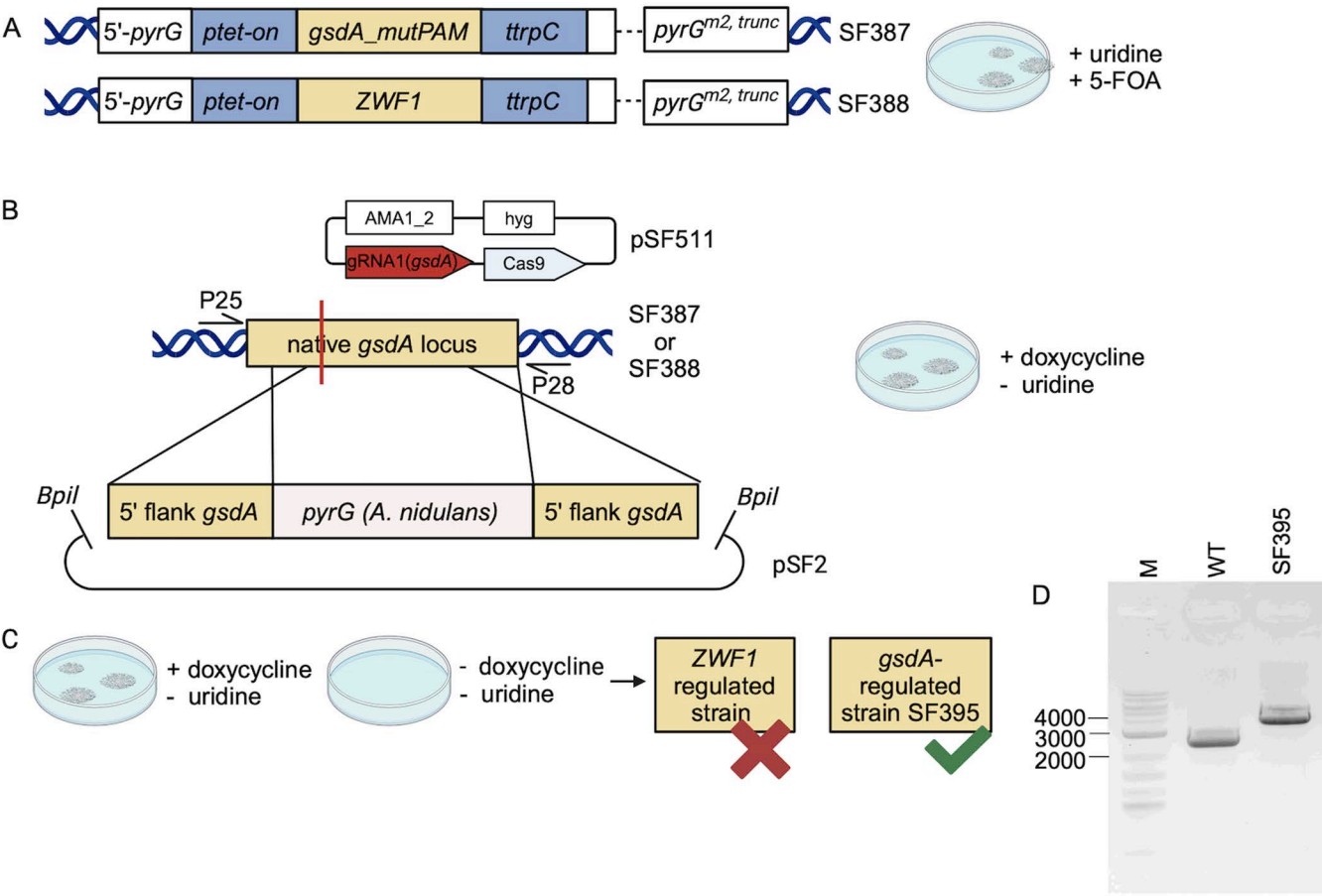

**Fig 3. Construction of a *gsdA*-regulated *A. niger* strain.** (A) The *gsdA* OE strain SF387 and *ZWF1* OE strain SF388 harbor the ptet-on regulated expression cassette at the *pyrG* locus. 5'-*pyrG* and the truncated version of *pyrG* (*pyrG*^m2,trunc) derived from respective transformation vectors for genomic integration via homologous recombination. Both strains carry an INDEL mutation in the *pyrG*^m2,trunc sequence of the *pyrG* locus (dotted line) and were selected on plates supplemented with uridine and 5-FOA. (B) SF387 and SF388 were transformed with a Cas9/sgRNA containing plasmid pSF511 designed for a double-strand break event at the native *gsdA* locus (red line). The co-transformed disruption cassette was linearized from plasmid pSF2 and harbored the *pyrG* gene of *A. nidulans* with ~1 kb of its up- and downstream sequence. Flanks of the 5' and 3' end *gsdA* gene were provided for homologous directed repair of the DNA lesion. Transformation was done on plates without uridine and supplementation of doxycycline to induce *gsdA* expression at the *pyrG* locus. Primers P25 and 28 were used for PCR verification of the integration event. (C) After transformation only the *gsdA* OE strain SF387 was uridine prototroph, did not grow without doxycycline and was referred to strain SF395. (D) Gel electrophoresis shows the expected fragment of 3899 bp compared to the wild-type with 2563 bp using primers P25 and P28. Created in BioRender. Fritsche, S. (2024) BioRender.com/i92i049.

cassette at the *pyrG* locus was induced with doxycycline. Only transformants with an integrated *pyrG* gene at the *gsdA* locus would grow on transformation plates without uridine (Fig 3B). Twenty transformants of each, the *gsdA* and *ZWF1* OE strains, were selected for further phenotypic analysis. All transformants of the *ZWF1* OE strain were able to grow without induction of the OE cassette. In contrast, five strains derived from the *gsdA* OE strain only grew on minimal medium with supplementation of doxycycline (Fig 3C). After single-conidia purification, four out of five transformants restored growth on plates without doxycycline and PCR analysis showed the native *gsdA* configuration of the locus. However, one strain, did only grow in the presence of doxycycline. PCR analysis confirmed the expected integration of the disruption cassette at the *gsdA* locus (Fig 3D) and sequence analysis confirmed the insertion of the *pyrG* sequence from *A. nidulans* seven base pairs upstream of the targeted site (S3 File).

## Phenotypic characterization of *A. niger* with regulated *gsdA* expression

During the screening of transformants for a disrupted *gsdA* locus we noticed that the resulting strain is not viable on glucose. Growth was only obtained on screening plates supplemented with doxycycline. Therefore, we continued to analyze the dependence of growth with different induction levels for the *gsdA* copy controlled by the ptet-on promoter. The spotting test shows a dilution series of conidia of the wild-type strain ATCC 1015 and the *gsdA*-regulated strain SF395 on glucose (Fig 4A). No growth was seen for SF395 without doxycycline addition and biomass formation was raising with increased concentration of doxycycline. The supplementation of 5 µg/mL doxycycline resulted in growth comparable to the wild-type strain but was

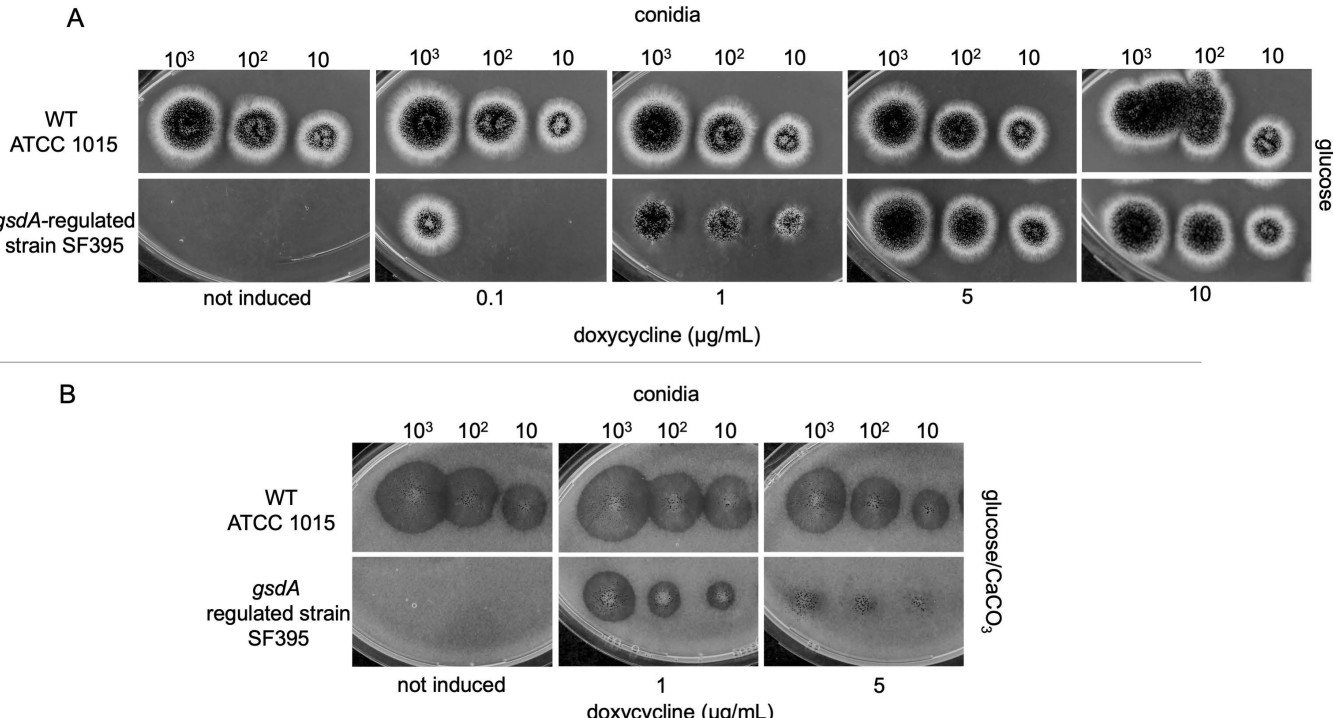

**Fig 4. Phenotypic characterization of *A. niger* with regulated *gsdA* expression.** Conidia of wild-type strain ATCC 1015 (WT) and the *gsdA*-regulated strain SF395 were grown on (A) minimal medium with 1% glucose to show dependency of growth on *gsdA* expression. (B) Strains were spotted on medium containing 20% glucose and $CaCO_3$ to screen for acidification of the $CaCO_3$. Increasing concentrations of doxycycline were applied to the media to regulate the expression of *gsdA* under the control of the tet-on promoter system in SF395. Growth plates were incubated at 30°C for 90h.

not further increased when induced with 10 µg/mL doxycycline. We next wanted to test how different induction levels influence organic acid production. The *gsdA*-regulated strain SF395 was spotted on medium composed of glucose and $CaCO_3$ (Fig 4B) and the clearing zone (halo) around the colony was evaluated as an indicator for acidification secreted. As expected, the *gsdA*-regulated strain is not growing on medium supplemented with glucose alone. However, at 1 µg/mL doxycycline growth can be observed together with a halo formation and hyphae are spread within the halo area. The back side of the growth plate is shown in S1 Fig. With an area of 1.21 cm² the halo is however smaller compared to the wild-type strain ATCC 1015 that measures 4.07 cm² at an inoculation density with 10² conidia per spot and which is not significantly influenced by the doxycycline level. With 5 µg/mL doxycycline the *gsdA*-regulated strain is growing, but no halo is formed under these conditions indicating a reduced acidification of the media, which is in accordance with the previous data (Fig 2) suggesting that increased *gsdA* expression leads to reduced citrate secretion.

We showed that on glucose the non-induced *gsdA*-regulated strain is dependent on the first reaction of the oxPPP which is the conversion of glucose 6-phosphate to 6-phosphogluconolactone. We assume that the strain can bypass this step by taking up gluconate, which is then channeled into the oxPPP via 6-phosphogluconate. To test this, the *gsdA*-regulated strain SF395 was spotted onto minimal medium with Na-gluconate as carbon source. Growth was compared to the control strain SF374, containing the native *gsdA* gene (Fig 5). The control strain was able to utilize Na-gluconate and doxycycline addition had no negative effect on growth. In contrast to glucose, as the sole carbon source, the *gsdA*-regulated strain SF395 could grow on Na-gluconate under non-inducing conditions. However, a minimum inoculation density of 10³ conidia/spot was required for outgrowth. Growth was increased with the addition of 1 µg/mL doxycycline, but stayed constant when up to 5 µg/mL doxycycline were supplemented.

## Regulating *gsdA* expression to influence citric acid production

We showed that the *gsdA*-regulated strain SF395 requires induction of the expression cassette with 0.1 µg/mL doxycycline to be able to grow. Furthermore, biomass production correlated

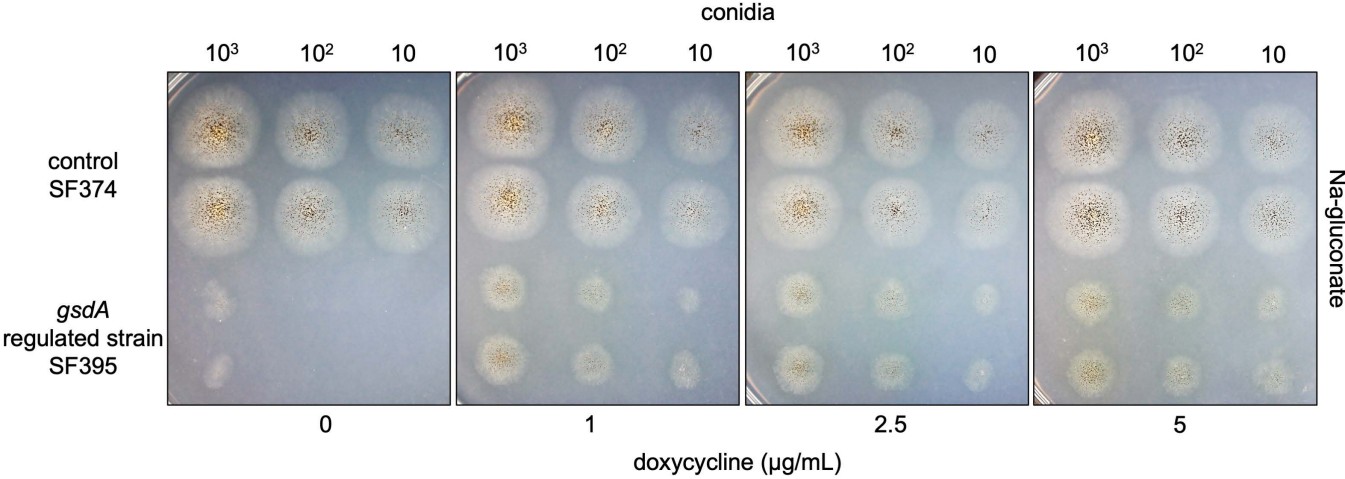

**Fig 5. Viability of *A. niger* on gluconate with regulation of *gsdA* expression levels.** The control strain SF374 and the *gsdA*-regulated strain SF395 were grown on a minimal medium with 1% Na-gluconate. Increasing concentrations of doxycycline were applied to the medium to regulate the expression of *gsdA* under the control of the ptet-on promoter system in SF395. Growth plates were incubated at 30°C for 90 h.

positively with increasing doxycycline levels. Expression of *gsdA* with 1 μg/mL doxycycline allowed acidification, while at 5 μg/mL, there is no indication for acidification of the medium. Conclusively, doxycycline levels ranging from 0 to 2.5 μg/mL were selected for shake flask cultivations and to monitor citric acid, erythritol and glycerol secretion (Fig 6). Generally, citric acid titers were lower in the *gsdA*-regulated strain SF395 compared to the control strain SF374. As shown on minimal medium, the *gsdA*-regulated strain SF395 was not able to grow without induction of the *gsdA* expression cassette. Hence, no titers of the target metabolites were detected in the supernatant of the culture. With 0.25 μg/mL doxycycline up to 4.24 ± 0.12 g/L citric acid was produced by the *gsdA*-regulated strain SF395. Citric acid titers were gradually decreasing with increased doxycycline concentrations, indicating the emphasized direction of glucose into the oxPPP (Fig 6A). Considering glucose consumption together with citric acid production, this condition resulted in a higher yield for citric acid on glucose for the *gsdA*-regulated strain SF395 compared to the control strain SF374 with 0.18 ± 0.02 and 0.12 ± 0.02 g/g, respectively (Fig 6B). There was no trend observed for erythritol titers but glycerol titers were increasing with increasing doxycycline concentration (Figs 6C and 6D).

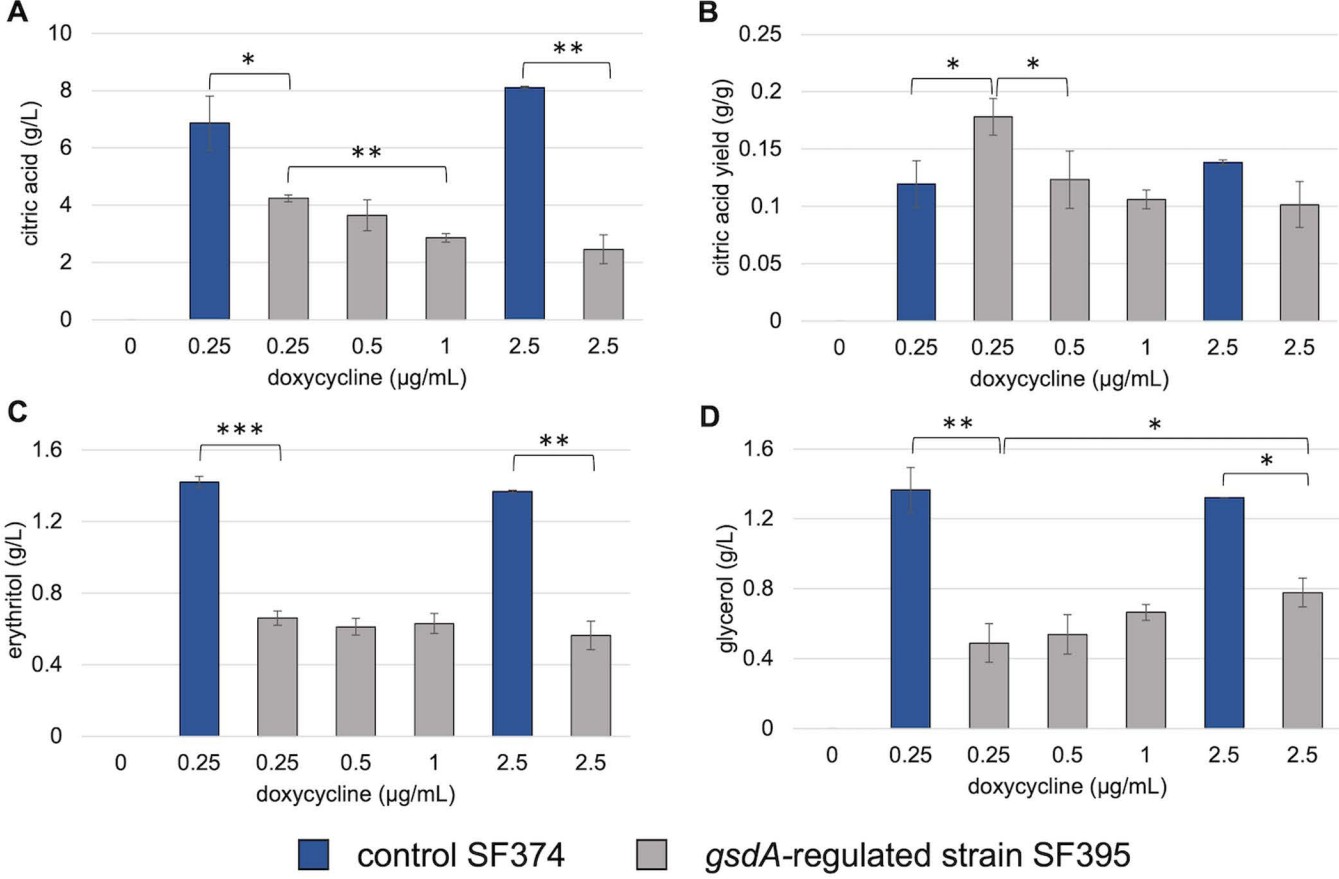

**Fig 6. *A. niger* with regulated *gsdA* expression cultivated in medium containing glucose.** $10^9$ conidia/L of the control strain SF374 or the *gsdA*-regulated strain SF395 were inoculated in Vogel's medium with 0–2.5 μg/mL doxycycline to induce *gsdA* under the tet-on promoter system at the *pyrG* locus in SF395. (A) Citric acid titer and (B) yield on glucose, (C) erythritol and (D) glycerol levels were measured after 168 h of cultivation at 30°C and 200 rpm. Error bars indicate standard deviation of three biological replicates. Significant differences (*p<0.05; **p<0.001; ***p<0.0001) were determined by t-test and high-lighted by asterisks.

Next, we hypothesized that the non-induced *gsdA*-regulated strain SF395 maintains growth through uptake of gluconate and directing it into the oxPPP. Through this, added glucose is then channeled to the glycolytic pathway and metabolized to citric acid more efficiently. To test this, Vogel's medium was supplemented with 20% sugar composed of glucose/gluconate ratios 80:20 and 60:40. Growth of the *gsdA*-regulated strain in shake flask cultivations was, however, severely affected and citric acid production could not be analyzed under these conditions. The medium was subsequently supplemented with 1 or 2.5 µg/mL doxycycline to compensate for this effect and citric acid production was measured over time ([Fig 7A]). The citric acid titers of the control strain SF374 reached around 50% higher titers in the medium with a glucose/gluconate ratio 80:20 compared to 60:40. In contrast, citric acid levels of the *gsdA*-regulated strain SF395 were similar in both conditions but were generally lower reaching after 168 hours only 5 g/L compared to 19 g/L of the control strain in the medium with a glucose/gluconate ratio 80:20 and 10 g/L with the ratio 60:20. As shown previously on solid medium, growth was constrained on glucose or Na-gluconate with induction of *gsdA* expression below 5 µg/mL doxycycline. Biomass production of the *gsdA*-regulated strain was also lower compared to the control strain in shake flask cultivations (S2 Fig). Citric acid yield on carbon source of the control strain increased gradually over time in both glucose/gluconate

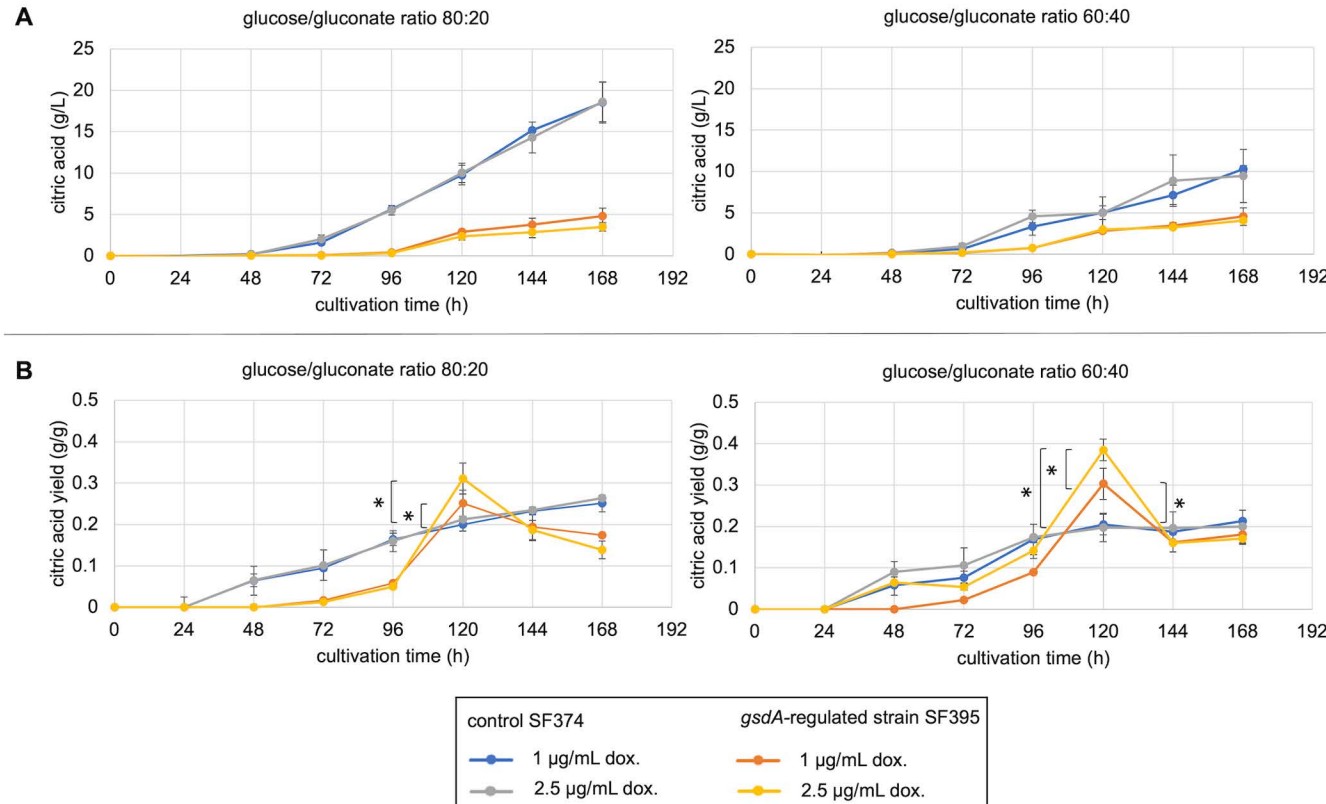

**Fig 7. *A. niger* with regulated *gsdA* expression cultivated in medium containing glucose and gluconate.** 10⁹ conidia/L of the control strain SF374 or the *gsdA*-regulated strain SF395 were inoculated in Vogel's medium with 1 and 2.5 µg/mL doxycycline (dox) to induce *gsdA* under the tet-on promoter system at the *pyrG* locus in SF395. Medium contained 20% (w/v) of a glucose/gluconate mixture with a ratio of either 80:20 or 60:40. (A) Citric acid titer and (B) citric acid yield on glucose/gluconate consumption measured in the supernatant of the culture during 168 h of cultivation at 30°C and 200 rpm. Error bars indicate standard deviation of three biological replicates. Asterisks in (B) mark significant differences ($p < 0.05$) of means from SF395 samples induced with 1 and 2.5 µg/mL doxycycline and were determined by t-test (n= 3).

mixtures (Fig 7B). This was in contrast to the results of the *gsdA*-regulated strain. Here, citric acid production was delayed and yields increased most between 96 and 120 hours before dropping to a lower level towards the end of cultivation. It is noteworthy that the yields of the control strain were exceeded by the *gsdA*-regulated strain after 120 hours of cultivation. This was observed for both media compositions, but was more significant for glucose/gluconate ratio 60:40. Furthermore, an increase of doxycycline from 1 to 2.5 µg/mL had a positive effect on the citric acid yield in the strain SF395 in both media.

Erythritol and glycerol titers of the *gsdA*-regulated strain SF395 and the control strain SF374 followed similar trends compared to the previously described citric acid results and are summarized in S3 Fig. In brief, more erythritol and glycerol was produced by SF374 in medium with a glucose/gluconate ratio of 80:20 compared to 60:40. The erythritol and glycerol production levels of the *gsdA*-regulated strain SF395 were similar in both conditions, but were generally lower compared to the control. In contrast to the observation of citric acid, erythritol and glycerol yields were not higher in the *gsdA*-regulated strain compared to the control strain after 120 h of cultivation. Generally, production of both metabolites was time-delayed in the strain SF395 and started after 144–168 h. At the end of the cultivation, the yield of erythritol was similar to that of the control strain in both media compositions. On the other hand, less glycerol was produced by the *gsdA*-regulated strain per g glucose/gluconate mixture consumed.

## Discussion

G6PD, encoded by the gene *gsdA* in *A. niger*, is a gateway for glucose to enter the PPP. The enzyme channels glucose-6-phosphate from glycolysis into the oxidative part of the PPP. This pathway is a main source of NADPH, which is essential for anabolic reactions. As in previous attempts to generate a *gsdA* deletion strain [42], such a strain was not obtained. Therefore, the approach of achieving tightly controlled expression of *gsdA* at the level of a single ptet-on regulated copy seems to be an attractive strategy to learn more about the essentiality of the *gsdA* gene in *A. niger*. In fact, it was challenging to obtain the replacement of *gsdA* gene at its native locus and thus first the effects of the upregulation of *gsdA* on citric acid secretion were studied. According to the classical model that glycolysis is required for citric acid secretion [43] it was observed that increased expression of *gsdA* reduces the secretion of citric acid. This can be explained by a higher flux through the PPP pathway, a lower flux in the TCA cycle and an increased biomass formation. A positive correlation between *gsdA* expression levels and biomass production has been previously reported, which was associated with reduced flux towards citrate and lower intracellular citrate levels [44]. We show that growth is dependent on the induction of ptet-on regulated *gsdA*. Interestingly erythritol and glycerol production was also reduced with higher *gsdA* induction levels, which indicates that an increased flux through the oxidative PPP is not the main cause for the formation of these by-products of citric acid production.

The effects of increased glucose entry into the PPP are potentially complex. G6PD is regulated by NADPH/NADP+ ratio and *in vitro* measurements showed that it is inhibited by high NADPH concentrations [45–47]. This feedback regulation could cause a limitation in overproducing metabolites of the PPP and hence, a limitation in biomass production. In addition, G6PD activity could also be limited through feedback inhibition of intermediates accumulated in the oxidative branch of the PPP. The concentration of 6-phosphogluconate has been reported to be high in strains overexpressing genes encoding for G6PD [25]. Notably, titers of 6-phosphogluconate further increased with overexpressing the gene for 6PGL, the second enzyme of the PPP. This indicated a bottleneck further downstream and hence, no increase

of the non-oxPPP metabolites [9]. In our study we showed that titers of erythritol indeed did not change significantly when *gsdA* was upregulated as a single copy via the inducible ptet-on promoter. Furthermore, erythritol titers were even decreasing with native *gsdA* expression in combination with overexpression of the second copy. This is in alignment with low erythritol levels that were previously reported in a strain overexpressing 6PGL [11].

Since increased *gsdA* concentrations resulted in decreased citric acid formation, it can be speculated that decreased *gsdA* concentrations may increase flux through glycolysis and thus increase citric acid excretion. However, the generation of a null mutant has failed in the past and was also not possible in this work despite the repeated application of advanced CRISPR/Cas9 genetic engineering methods.

In this study, we generated an *A. niger* strain overexpressing a second copy of *gsdA* via the inducible ptet-on system at the *pyrG* locus. Subsequently, we disrupted the native *gsdA* locus and were able to regulate its expression via the inducible promoter. The tight regulation of this system is shown by the fact that the strain can no longer grow on glucose-containing plates without the inducer, whereas on plates with > 5 μg/mL it grows again like the control strain. To analyze the role of G6PD for viability, deletion studies were reported for organisms including *S. cerevisiae* [24,48] and the filamentous fungus *N. crassa* [49]. Generation of a *gsdA* deficient *A. niger* strain was yet unsuccessful. In fact, there is no report since the attempt of van den Broek et al., 1995 [42]. Their approach was designed to insert a disrupted version of the *gsdA* gene via homologous recombination using hygromycin as a selection marker [42]. In this study, we followed the strategy to disrupt the native *gsdA* gene while inducing a second copy at *the pyrG* locus. The resulting strain did not grow on glucose under non-inducing conditions. Growth was restored when gluconate was provided as a carbon source and indicates that gluconate is taken up and enters the PPP after conversion to 6-phosphogluconate. Interestingly, no growth is obtained in the *gsdA*-regulated strain on glucose alone although *A. niger* has the ability to oxidize glucose to gluconate. However, the uptake and the conversion to 6-phosphogluconate might be under carbon catabolite repression. This hypothesis is supported by previous data showing that transcription of the gluconokinase on glucose, GukA, is repressed in the wild-type strain compared to a *creA* knock-out mutant (log2FC: -3,4; Adj. P-value: 0,004) [49]. Notably, a minimum of conidia of the non-induced *gsdA*-regulated strain was required to establish mycelial growth on gluconate and growth was significantly delayed compared to the control strain. The transport of gluconate into the cell or the phosphorylation via gluconokinase could be a possible bottleneck for the *gsdA*-regulated strain under non-inducing condition. However, the reaction of gluconokinase is ATP- and not NADPH dependent [50].

Another interesting observation was that upregulation of the inducible copy of *gsdA* did not compensate for the growth-delayed phenotype on gluconate. This suggests that in contrast to the control strain glucose-6-phosphate is not sufficiently provided via gluconeogenesis.

Downregulation of *gsdA* was of particular interest to alter citric acid production in the shake flask culture of *A. niger*. Although we show that more citric acid can be produced from glucose at a lower expression of *gsdA*, the production capacity is restricted to a limited time window due to the interdependence of biomass and citric acid production. We anticipated to support biomass production in the *gsdA* downregulated strain by supplementation of gluconate and channeling it as 6-phospholuconate into the PPP. We hypothesized, that the glucose in the medium is then metabolized through the glycolytic pathway more efficiently to produce citric acid. We indeed observed higher yields of citric acid during the cultivation compared to the parental strain. In addition, this was improved with increased availability of gluconate. Interestingly, the improved yield reached its peak after 120 h but in a later stage of the cultivation dropped to lower levels. One explanation could be that continuous expression of *gsdA*

through doxycycline is different to the regulation of the gene under native conditions. As a result, a metabolic imbalance could lead to less citric acid production.

## Conclusion

G6PD is a crucial metabolic step in glucose metabolism and enables the operation of the oxidative part of the PPP. Conclusively, a strain with a diminished expression of *gsdA* gene is not growing on glucose as the sole carbon source, but can grow on gluconate. With the strategy presented, the interaction between growth and citric acid production could be analyzed by regulating expression of *gsdA*. Increased *gsdA* expression significantly reduced citrate secretion. Using the ptet-on expression system, it was possible to downregulate *gsdA* expression and it was found that with lower *gsdA* expression, increased citric acid accumulation is possible. However, this competes with a simultaneous reduction in biomass formation and can therefore only be achieved in a limited time window during a batch process. Nevertheless, the ability to uncouple the expression of *gsdA* from its native promoter allows a significant intervention in the glucose metabolism of *A. niger* and may be a valuable tool to study further reactions of the pentose phosphate pathway in isolation.

## Supporting information

**S1 File. Coding sequence of *gsdA_mutPAM* in pSF8 and pSF503.**
(PDF)

**S2 File. Result of Sanger sequencing the *pyrG* locus of SF387 and SF388 using primer P23.**
(PDF)

**S3 File. Result of Sanger sequencing the native *gsdA* locus from SF395 using primers P25 to P28.**
(PDF)

**S1 Fig. Acidification of $CaCO_3$ of *A. niger* with regulated *gsdA* expression.** Back side of growth plate presented in Fig 4B showing hyphae spread in the area of the halo. Increasing concentrations of doxycycline were applied to the media to regulate the expression of *gsdA* under the control of the tet-on promoter system in SF395. Growth plates were incubated at 30°C for 90h.
(JPG)

**S2 Fig. *A. niger* with regulated *gsdA* expression cultivated in medium containing glucose and gluconate.** $10^9$ conidia/L of the control strain SF374 or the *gsdA*-regulated strain SF395 were cultivated in shake flasks with Vogel's medium with 1 and 2.5 μg/mL doxycycline to induce *gsdA* under the ptet-on promoter system at the *pyrG* locus in SF395. Medium contained 20% (w/v) of a glucose/gluconate mixture with a ratio of either (A) 80:20 or (B) 60:40. Images were taken after 120 h of cultivation at 30°C and 200 rpm.
(JPG)

**S3 Fig. Side product formation and C-source consumption of *A. niger*** with regulated *gsdA* expression cultivated in medium containing glucose and gluconate. 109 conidia/L of the control strain SF374 or the gsdA-regulated strain SF395 were inoculated in Vogel's medium with 1 and 2.5 μg/mL doxycycline to induce gsdA under the ptet-on promoter system at the pyrG locus in SF395. Medium contained 20% (w/v) of a glucose/gluconate mixture with a ratio of either 80:20 or 60:40. (A) Erythritol titers and (B) respective yields on consumed carbon source (C-source), (C) glycerol titers and (D) respective yields and (E) C-source consumption

were measured in the supernatant of the culture during 168 h of cultivation at 30°C and 200 rpm.
(JPG)

**S1 raw image. This gel is the uncropped image that was used to generate** Fig 3 **of the results.** The gel image was generated with a Gel Doc XR+ Gel Documentation System.
(PDF)

## Acknowledgments

We highly appreciate the work of Michael Pesek who helped to generate and screen SF395 mutants and who helped to perform the spotting tests. Furthermore, we want to thank Felix Fronek and Henrich Novotny for the support in the lab.

## Author contributions

**Conceptualization:** Susanne Fritsche, Matthias G. Steiger.

**Data curation:** Susanne Fritsche, Matthias G. Steiger.

**Formal analysis:** Susanne Fritsche.

**Investigation:** Susanne Fritsche, Güler Demirbas-Uzel, Matthias G. Steiger.

**Methodology:** Susanne Fritsche, Valeria Ellena, Güler Demirbas-Uzel, Matthias G. Steiger.

**Project administration:** Matthias G. Steiger.

**Resources:** Matthias G. Steiger.

**Supervision:** Matthias G. Steiger.

**Visualization:** Susanne Fritsche.

**Writing – original draft:** Susanne Fritsche.

**Writing – review & editing:** Susanne Fritsche, Valeria Ellena, Güler Demirbas-Uzel, Matthias G. Steiger.

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
