## [Decision Letter · Decision Letter 0]

24 Oct 2024

PONE-D-24-42906Regulating the glucose-6-phosphate dehydrogenase encoding gene gsdA and its impact on growth and citric acid production in Aspergillus nigerPLOS ONE

Dear Dr. Steiger,

Thank you for submitting your manuscript to PLOS ONE. After careful consideration, we feel that it has merit but does not fully meet PLOS ONE’s publication criteria as it currently stands. Therefore, we invite you to submit a revised version of the manuscript that addresses the points raised during the review process.

We look forward to receiving your revised manuscript.

Kind regards,

Bashir Sajo Mienda, PhD

Academic Editor

PLOS ONE

Reviewers' comments:

Reviewer's Responses to Questions

**Comments to the Author**

1. Is the manuscript technically sound, and do the data support the conclusions?

Reviewer #1: Partly

Reviewer #2: Yes

2. Has the statistical analysis been performed appropriately and rigorously? 

Reviewer #1: No

Reviewer #2: Yes

3. Have the authors made all data underlying the findings in their manuscript fully available?

Reviewer #1: Yes

Reviewer #2: Yes

4. Is the manuscript presented in an intelligible fashion and written in standard English?

Reviewer #1: Yes

Reviewer #2: Yes

5. Review Comments to the Author

Reviewer #1: The authors studied to assess the effects of varying the gsdA expression levels in A. niger. While some studies in A. niger had analyzed the impact of varying the gsdA expression levels on NADPH levels and glucoamylase production in A. niger, in this study, the authors studied the effect of overexpressing gsdA on citric acid production in A. niger.

This manuscript includes interesting lines that should be studied, while this is preliminary to be accepted.

First of all, all data need to be statistically analyzed.

When gsdA is expressed in a tet-on system, the authors need to quantify the gsdA expression levels and NADPH levels in transformants.

Ln238, Fig. 2B, Please describe that how to messure the glucose concentration.

Fig. 2b, c, and d legends do not match the figure.

Fig. 4b It is hard to understand from the image whether the basal hyphae were spreading or a halo. High-resolution pictures are needed.

Can it be said that the lack of halo with adding 5 µg doxycycline is linked to citric acid production? Other organic acid production also needs to be quantified.

Fig. 7b Is the increase in citric acid production versus the concentration of carbon sources between 96-120 h reflecting the reduction of available carbon sources?

The authors also need to describe how to quantify carbon sources.

Minor comments

Ln63-65

Please describe how much ethanol production has improved in S. cerevisiae.

Ln621, DOI may be this, “10.1007/s00253-012-4207-9”.

Reviewer #2: I reviewed the article entitled “Regulating the glucose-6-phosphate dehydrogenase encoding gene gsdA and its impact on growth and citric acid production in Aspergillus niger”.

The authors presented an A. niger strain expressing a ptet-on regulated version of gsdA at the pyrG locus.

The manuscript was beautifully planned and realized. The work is interesting, informative, revealing and deals with an important area of research in the field of PLOS ONE. But I have some suggestions:

1. In Abstract: The statement “..the split ratio of carbon between glycolysis and the pentose phosphate pathway” is not suitable for metabolism. I think the statement "It determines which metabolic pathway glucose 6-phosphate will go to" would be more appropriate.

2. In Conclusion: It should be written as "G6PD is a crucial metabolic step in central glucose metabolism" instead of "G6PD is a crucial metabolic step in central carbon metabolism".

3. In all parts of the article:: It should be written as " glucose metabolism" instead of " carbon metabolism". Because other monosaccharides are involved in the metabolic pathway that glucose enters. Again, the carbon source in metabolism is not only monosaccharides. Fatty acids and amino acids also contain carbon.

4. It is not preferred for primers to have the same bases one after the other. Some primers show the order TTTT, CCCC. How did this affect the results?

6. PLOS authors have the option to publish the peer review history of their article (what does this mean? ). If published, this will include your full peer review and any attached files.

**Do you want your identity to be public for this peer review?** For information about this choice, including consent withdrawal, please see our Privacy Policy .

Reviewer #1: No

Reviewer #2: **Yes: ** Ömer İrfan Küfrevioğlu

---

## [Author Response · Author response to Decision Letter 1]

7 Feb 2025

Dear Editor,

We would like to thank both reviewers for the overall positive assessment of the manuscript and the constructive feedback.

As a main change we measured G6PD in control and an engineered strain and with this new data are able to confirm that our overexpression constructs are working as expected. Furthermore, we implemented the suggested comments and corrected errors and are now able to provide an im-proved version of the manuscript containing new data, which we hope will get your approval.

Changes to the manuscript are in blue color.

In the uploaded point-bypointreply you will find our detailed responses to the reviewers' comments.

With kind regards,

Matthias Steiger

---

## [Decision Letter · Decision Letter 1]

5 Mar 2025

Regulating the glucose-6-phosphate dehydrogenase encoding gene gsdA and its impact on growth and citric acid production in Aspergillus niger

PONE-D-24-42906R1

Dear Dr. STEIGER,

We’re pleased to inform you that your manuscript has been judged scientifically suitable for publication and will be formally accepted for publication once it meets all outstanding technical requirements.

Kind regards,

Bashir Sajo Mienda, PhD

Academic Editor

PLOS ONE

Additional Editor Comments (optional):

Reviewers' comments:

Reviewer's Responses to Questions

**Comments to the Author**

1. If the authors have adequately addressed your comments raised in a previous round of review and you feel that this manuscript is now acceptable for publication, you may indicate that here to bypass the “Comments to the Author” section, enter your conflict of interest statement in the “Confidential to Editor” section, and submit your "Accept" recommendation.

Reviewer #1: All comments have been addressed

2. Is the manuscript technically sound, and do the data support the conclusions?

Reviewer #1: Yes

3. Has the statistical analysis been performed appropriately and rigorously? 

Reviewer #1: Yes

4. Have the authors made all data underlying the findings in their manuscript fully available?

Reviewer #1: Yes

5. Is the manuscript presented in an intelligible fashion and written in standard English?

Reviewer #1: Yes

6. Review Comments to the Author

Reviewer #1: (No Response)

7. PLOS authors have the option to publish the peer review history of their article (what does this mean? ). If published, this will include your full peer review and any attached files.

**Do you want your identity to be public for this peer review?** For information about this choice, including consent withdrawal, please see our Privacy Policy .

Reviewer #1: No

---

## [Editor Report · Acceptance letter]

PONE-D-24-42906R1

PLOS ONE

Dear Dr. Steiger,

I'm pleased to inform you that your manuscript has been deemed suitable for publication in PLOS ONE. Congratulations! Your manuscript is now being handed over to our production team.

Kind regards,

on behalf of

Dr. Bashir Sajo Mienda

Academic Editor

PLOS ONE